# CLUSTERLLM: Large Language Models as a Guide for Text Clustering

**Yuwei Zhang    Zihan Wang    Jingbo Shang**[*]
University of California, San Diego
{yuz163, ziw224, jshang}@ucsd.edu

## Abstract

We introduce CLUSTERLLM, a novel text clustering framework that leverages feedback from an instruction-tuned large language model, such as ChatGPT. Compared with traditional unsupervised methods that builds upon "small" embedders, CLUSTERLLM exhibits two intriguing advantages: (1) it enjoys the emergent capability of LLM even if its embeddings are inaccessible; and (2) it understands the user's preference on clustering through textual instruction and/or a few annotated data. First, we prompt ChatGPT for insights on clustering perspective by constructing hard triplet questions <*does A better correspond to B than C*>, where *A*, *B* and *C* are similar data points that belong to different clusters according to small embedder. We empirically show that this strategy is both effective for fine-tuning small embedder and cost-efficient to query ChatGPT. Second, we prompt ChatGPT for helps on clustering granularity by carefully designed pairwise questions <*do A and B belong to the same category*>, and tune the granularity from cluster hierarchies that is the most consistent with the ChatGPT answers. Extensive experiments on 14 datasets show that CLUSTERLLM consistently improves clustering quality, at an average cost of ∼$0.6[1] per dataset. The code will be available at https://github.com/zhang-yu-wei/ClusterLLM.

## 1 Introduction

Text clustering, as a fundamental task in natural language processing (NLP), has a wide spectrum of applications, such as identifying public perception from social media (Park et al., 2022), analysing cause of accidents (Xu et al., 2022), and detecting emerging research topics (Martínez et al., 2022). A common practice for text clustering is to apply clustering algorithms (MacQueen, 1967; Zhang et al.,

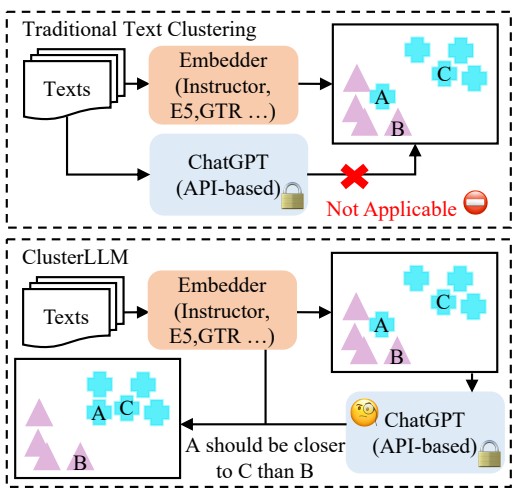

Figure 1: LLMs like ChatGPT are not applicable for text clustering directly because of the inaccessible embeddings. CLUSTERLLM resolves the dilemma by leveraging LLM as a guide on text clustering.

2021a) on top of pre-trained embedders (Muennighoff et al., 2022; Wang et al., 2022; Su et al., 2022) which could achieve higher performance with better pre-training quality. State-of-the-art large language models (LLMs) such as recent GPT series (Brown et al., 2020; Ouyang et al., 2022; OpenAI, 2023) have demonstrated extraordinary language capabilities for various NLP applications however, these GPT models can only be utilized through the APIs without accessible embedding vectors for clustering. Hence, LLMs cannot be directly applied on text clustering tasks.

In this paper, we provide insights on the question: *Can we leverage API-based LLMs to guide text clustering efficiently?* We attack this challenging question by drawing inspiration from an observation that *humans represent an instance through comparing with others* (Nosofsky, 2011). For instance, people often classify a new piece of music into a specific genre by relating to familiar ones. In fact, pairwise relationships have been utilized in spectral clustering (Donath and Hoffman, 1972; Cheeger, 1970) before. Nonetheless, naively

---

[*] Corresponding author.
[1]The cost is calculated with gpt-3.5-turbo.

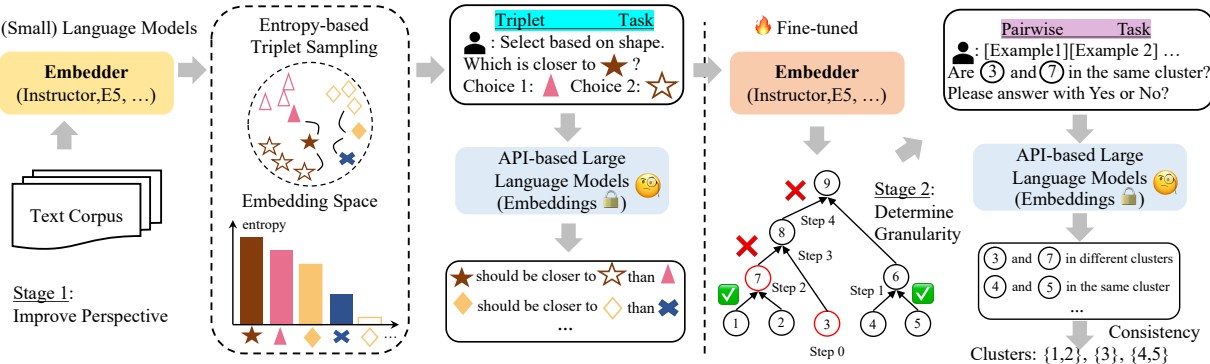

Figure 2: An overview of CLUSTERLLM. It utilizes LLM to guide an embedder for text clustering with a low cost.

traversing all the pairs within dataset is obviously intractable and too expensive for querying LLMs.

We propose CLUSTERLLM, a framework that utilizes LLM to guide a small embedder for finding text clusters with a low cost, as shown in Figure 1. It comprises two stages that are specially designed for two aspects of clustering: (1) perspective, i.e., the grouping criterion such as *topic*, *intent* and *emotion* and (2) granularity, i.e. the scope of clusters.

In Stage 1, we prompt LLMs with a triplet task that predicts which one of the two candidate choices is closer to anchor instance to understand the user-preferred perspectives. We choose this triplet task because (a) it is irrelevant with cluster granularity and (b) the produced triplets can fine-tune small embedder towards the right perspective. In order to improve sample efficiency, we further propose entropy-based triplet sampling to find the most informative triplets. Specifically, we first calculate entropy for each instance based on cluster assignment probabilities, and then identify those with highest entropy. Two candidate choices are then sampled from its nearest clusters to guarantee they are close enough to the anchor.

In Stage 2, we first obtain the cluster hierarchy that starts from instance-level clusters and iteratively merge two closest clusters until the entire dataset. And then we prompt LLMs to determine cluster granularity with a few annotated data pairs as demonstrations. We construct the data pairs to prompt by sampling from two clusters that are merged at each step of hierarchical clustering, so that they cover a wide range of granularities. And the final decision is made by measuring consistency between each level of clustering and predictions.

We extensively evaluate CLUSTERLLM on 14 datasets that include diverse tasks such as intent discovery, topic mining, type discovery, domain discovery, and emotion detection. Furthermore, these datasets span a wide range of granularities that have 10 to 150 number of clusters. We show that CLUSTERLLM is effective overall on improving clustering quality, where the clustering performance is improved over both a deep clustering baseline and a self-supervise baseline. Moreover, the ablation study shows that our sampling strategy is effective compared to a random sampling baseline. Finally, CLUSTERLLM also outperforms clustering-error based methods on determining cluster granularity.

In summary, our contributions are three-fold: (i) We propose a framework CLUSTERLLM that utilizes sentence relations predicted from API-based LLMs to guide clustering. Furthermore, it allows users to provide textual instructions and/or few-shot annotations to specify preferences on clustering. (ii) In order to reduce API-queries, we propose a novel entropy-based sampling strategy to find the most informative triplets. Additionally, we utilize pairwise data sampled from hierarchical clustering to determine cluster granularity. (iii) Extensive experiments show that our proposed method can improve clustering performance at ~$0.2 for perspective and ~$0.4 for granularity with GPT-3.5.

## 2 Preliminary

Text clustering takes an unlabeled corpus $\mathcal{D} = \{x_i\}_{i=1}^N$ as input, and outputs a clustering assignment $\mathcal{Y} = \{y_i\}_{i=1}^N$ that maps the input text to cluster indices. To specify user's needs, CLUSTERLLM integrates additional textual instruction (e.g. "Select the example that better corresponds with the Query in terms of entity type.") to understand perspective and few-shot annotations (e.g. "Sentence1 and Sentence2 have the same entity type ...") to determine cluster granularity.

## 3 Our CLUSTERLLM

CLUSTERLLM is based on a pre-trained small embedder (Wang et al., 2022; Su et al., 2022) (denoted

as $f$) which usually represents sentences individually. In contrast, inspired by human cognitive ability (Nosofsky, 2011), CLUSTERLLM considers a pair or a triplet of sentences through prompting LLMs that are trained to follow human instructions (Ouyang et al., 2022; OpenAI, 2023). Specifically, CLUSTERLLM is a two-stage framework (See Figure 2). In Section 3.1 we introduce Stage 1 that utilizes triplet task to improve clustering quality with respect to user-specified perspectives, along with a sampling strategy that reduces number of API queries. In Section 3.2, we introduce Stage 2 that leverages pairwise task to determine cluster granularity based on predictions from LLMs.

## 3.1 Triplet Task for Perspective

In this section, we explore how to harness a triplet task to refine the cluster structures for a user-specified perspective. A triplet task takes as input a tuple of three sentences $t = (a, c_1, c_2)$, where $a$ is the anchor and $(c_1, c_2)$ are two choices. We then prompt LLMs to select one of $(c_1, c_2)$ that better corresponds with $a$ using a prompt $\mathcal{P}_T$. Moreover, in order to specify the user's perspective, $\mathcal{P}_T$ also requires a task instruction $\mathcal{I}_T$ as input. The LLM should make a choice

$$c_j = \mathcal{P}_T(\mathcal{I}_T, t), \tag{1}$$

where $c_j \in \{c_1, c_2\}$ indicates one of the choices that LLM selects as positive and we denote the other (or negative) one as $c_{\setminus j}$.

### 3.1.1 Entropy-based Triplet Sampling

While one can randomly sample triplets to query the LLM, we demonstrate it non-efficient in experiments. In this section, we pose the question of mining *informative* triplets to both save the costs from querying LLMs and optimally improve the clustering. To achieve this, we resort to the current clustering results from the extracted embeddings $\mathcal{Z} = \{z_i = f(x_i)\}_{i=1}^N$. In summary, our algorithm contains two steps: **Step 1:** We find the most ambiguous instances as anchors based on entropy. **Step 2:** For each anchor instance, we sample two choices from two of its closest clusters. Refer to Algorithm 1 for entire process.

In Step 1, since the granularity is unknown at current stage, we perform clustering on top of $\mathcal{Z}$, where the clustering hyperparameters[2] are consis-

---

[2] It can be number of clusters in K-means or maximum distance to merge two clusters in hierarchical clustering

---

**Algorithm 1:** Entropy-based Triplet Sampling

**Input:** embeddings $\mathcal{Z} = \{z_i = f(x_i)\}_{i=1}^N$, interval boundaries $\gamma_{\text{high}}$ and $\gamma_{\text{low}}$, closest clusters fraction $\epsilon = 2\%$, maximum number of queries $Q$.

1  **Step 1:** $K$ clusters $\leftarrow$ *Clustering*($\mathcal{Z}$);
2  Compute $\mu_k$ for each cluster $k$ by averaging;
3  Compute $K_{\text{closest}}$ with Eq. 3;
4  Compute entropy $H$ with Eq. 4;
5  $Ind \leftarrow argsort(H)[::-1]$;
6  $Ind \leftarrow Ind[\gamma_{\text{high}}N:\gamma_{\text{low}}N]$;
7  **Step 2:** Initialize triplets $\{t_q\} \leftarrow \{\}$;
8  **while** $len(\{t_q\}) < Q$ **do**
9     **for** $a$ *in Ind* **do**
10       Obtain $K_{\text{closest}}$ closest clusters;
11       Sample $C_1, C_2$ from closest clusters;
12       $c_1 \sim C_1, c_2 \sim C_2, t = (a, c_1, c_2)$;
13       **if** $t$ *not in* $\{t_q\}, c_1 \neq a, c_2 \neq a$ **then**
14          Append $t$ to $\{t_q\}$;

**Output:** A set of triplets $\{t_q\}_{q=1}^Q$

---

tent across datasets and only specific to the embedder model $f$. Cluster center $\mu_k$ will thereafter be calculated for cluster $k$ by averaging embeddings assigned to it. Following (Xie et al., 2016; Van der Maaten and Hinton, 2008), we calculate instance-wise soft assignments with Student's $t$-distribution,

$$p_{ik} = \frac{(1 + ||z_i - \mu_k||^2/\alpha)^{-\frac{\alpha+1}{2}}}{\sum_{k'}(1 + ||z_i - \mu_{k'}||^2/\alpha)^{-\frac{\alpha+1}{2}}} \tag{2}$$

where $\alpha = 1$ is the degree of freedom. We then define closest clusters for instance $i$ as $K_{\text{closest}}$ clusters with largest soft assignment $p_{ik}$. Here, $K_{\text{closest}}$ is proportional to the total number of clusters $K$.

$$K_{\text{closest}} = \max(\epsilon K, 2) \tag{3}$$

where we fix $\epsilon$ to be a small value, such as $2\%$. We then compute entropy based on these closest clusters with renormalized probabilities $p'_{ik}$,

$$h_i = -\sum_{k=1}^{K_{\text{closest}}} p'_{ik} \log(p'_{ik}) \tag{4}$$

where $p'_{ik} = \frac{p_{ik}}{\sum_{k'=1}^{K_{\text{closest}}} p_{ik'}}$. We sort the entire dataset in descending order according to the entropies $H = \{h_i\}_{i=1}^N$. We introduce two hyperparameters $\gamma_{\text{high}}$ and $\gamma_{\text{low}}$ that control the proportion interval to filter out from ordered dataset. Our hypothesis is that higher entropy (smaller $\gamma_{\text{high}}$ and $\gamma_{\text{low}}$) anchors form more informative triplets that we verify in Section 4.6. In Step 2, we randomly sample two clusters $C_1, C_2$ from $K_{\text{closest}}$ closest clusters,

and then sample two sentences $c_1, c_2$ from each of them as choices (see line 11 and line 12). In another word, these choices should be either a positive or a hard negative to the anchor. Finally, we also remove triplets that are either repeated or have identical choice and anchor. We continue to sample triplets until reaching budget $Q$.

**Remarks.** (1) Since $Q$ is defined by the user and is independent with the dataset size, our sampling is cost-efficient. For example, in our experiments, using $1,024$ queries can improve performance on both dataset scales of $\sim 3,000$ and $\sim 50,000$. (2) From the view of ground truth, the sampled triplets might contain "both are correct" or "none of the above". However, we argue that even these triplets might provide soft aligning information, i.e. the ranking of closeness between choices. (3) Our sampling method may also be utilized in active learning to acquire human annotations when no prior knowledge is available on the categories.

### 3.1.2 Finetuning Embedder

Now that we have the triplet predictions, it is still not clear how to utilize them in clustering. Previous research rely on deep constrained clustering (Zhang et al., 2020; Manduchi et al., 2021) which are often sensitive to noisy labels (Basu et al., 2008). In this paper, we instead focus on finetuning the base embedder $f$ towards producing an embedding space that better explains the user's perspective. We exploit both hard and in-batch negatives. Following (Su et al., 2022; Ni et al., 2022b), for a triplet $t = (a, c_j, c_{\setminus j})$ with positive $c_j$ and hard negative $c_{\setminus j}$, we optimize the following objective,

$$l_j = \frac{\exp\left(s(a, c_j)/\tau\right)}{\sum_{c_l \in \mathcal{B}} \exp\left(s(a, c_l)/\tau\right)} \quad (5)$$

where $\mathcal{B}$ combines $c_j$, $c_{\setminus j}$ and other in-batch negatives. $\tau$ is a temperature parameter. Following the original implementation, we also compute the loss with $a$ and $c_j$ swapped. Finally fine-tuned embedders can be applied to find even more informative triplets with our sampling method which will further improve performance in an iterative manner. We acquire clustering assignments by running clustering algorithms on top of extracted embeddings.

### 3.2 Pairwise Task for Granularity

In this section, we build upon the refined embedding space in Section 3.1 to determine cluster granularity. In this paper, we convert the problem of determining granularity into finding the best step in a cluster hierarchy (see Figure 2 right), where each step denotes a unique granularity (or equally number of clusters). It is non-trivial, since different granularities can be applied to the same dataset (such as domains or topics). To tackle this challenge, we query LLM with pairwise task that predicts whether a pair of data $p$ belong to the same cluster with a prompt $\mathcal{P}_P$,

$$w = \mathcal{P}_P(\mathcal{I}_P, \{\tilde{p}_d\}_{d=1}^D, p) \quad (6)$$

where $w \in \{\text{same}, \text{different}\}$ is the binary decision, $\mathcal{I}_P$ is the task instruction and $\{\tilde{p}_d\}_{d=1}^D$ are few-shot demonstration pairs used for in-context learning (typically $D = 4$). We assume these demonstration pairs are annotated by users who have a desired cluster granularity in mind. We also combine a brief justification for each demonstration pair (see Table 12 bottom for example).

#### 3.2.1 Determine Granularity with Pairwise Hierarchical Sampling

We then introduce how to sample pairs from cluster hierarchy to query LLMs and determine granularity. We assume a maximum and a minimum number of clusters (denoted as $k_{\max}$ and $k_{\min}$) following Pelleg et al. (2000) which depend on the user's expectation on the granularity. We then randomly sample $\lambda$ (1 or 3 in our experiments) pairs of data from the two clusters to be merged at each step to form candidate pairs $\{p_i\}_{i=1}^{N_p}$, where $N_p = \lambda(k_{\max} - k_{\min})$. These pairs cover the entire range of granularity between $k_{\max}$ and $k_{\min}$, and will be used to query LLMs. After that, each level of granularity can be examined against LLM predictions to choose the one with the highest consistency measure $\mathcal{M}$,

$$k^* = \underset{k}{\operatorname{argmax}} \, \mathcal{M}(W^p, W^k) \quad (7)$$

where $W^p = \{w_i^p\}_{i=1}^{N_p}$ denotes the predictions obtained from Eq. 6 and $W^k$ represents a set of binary values indicating whether each pair of data is in the same cluster at granularity $k$. Empirically, we found the following performs better in our framework: use F-beta score, a weighted harmonic mean of precision and recall, as measurement $\mathcal{M}$ and set $W^p/W^k$ as labels/predictions. Finally, for large-scale datasets, we address the high time complexity of hierarchical clustering by applying it on top of mini-batch K-means. See details in Appendix A.

**Remarks.** Similar to Section 3.1.1, pairwise hierarchical sampling can also be used to acquire human

annotations. Nonetheless, the reliability of the algorithm still depends on the quality of clusters. In an extreme case where the clusters are completely random, it is unable to find granularity even though all the pairwise predictions are correct.

# 4 Experiments

We first evaluate CLUSTERLLM on clustering quality with ground truth number of clusters in Section 4.4. Then we conduct ablation studies in Section 4.6 to further analyze the effectiveness of CLUSTERLLM. Finally, we show results of determining cluster granularity in Section 4.7.

## 4.1 Datasets

We provide a high-level summary of evaluated datasets in this section, and see Appendix E for more descriptions. In this paper, we evaluate on a broad range of clustering datasets with various perspectives and granularities. Furthermore, to better analyze the effect of scale, each dataset has both a small-scale and a large-scale version. The two versions are different in number of data while keeping the same number of clusters. A summary of dataset statistics is shown in Table 1. Note that there is no data splits in clustering.

**Intent (Domain) Discovery.** Intent discovery (Zhang et al., 2021b, 2022) discovers unknown intents in unlabeled customer utterances. For CLINC, Massive and MTOP, we also use domains as labels to convert them into domain discovery.

**Type Discovery.** Type Discovery (Li et al., 2022) resolves the closed-world set-up of traditional Information Extraction. In this work, we focus on three tasks: entity, relation and event type discovery. To indicate specific mentions (entities or event triggers), we directly append them behind sentences with natural language formats, such as "The relation between [ENTITY1] and [ENTITY2]".

**Topic Mining.** We adapt three topic mining datasets from MTEB (Muennighoff et al., 2022).

**Emotion.** We adapt GoEmo (Demszky et al., 2020), a fine-grained emotion detection dataset by removing multi-label or neutral instances.

## 4.2 Experiment Details

**Query LLMs.** The prompt only contains a task-specific instruction (see Table 11). We set generation temperature to $0.5$. Explanations are suppressed by adding a postfix:"Please respond with 'Choice 1' or 'Choice 2' without explanation" and

| Task | Name | #clusters | #data(small) | #data(large) |
|---|---|---|---|---|
| Intent | Bank77 | 77 | 3,080 | 10,003 |
| | CLINC(I) | 150 | 4,500 | 15,000 |
| | MTOP(I) | 102 | 4,386 | 15,638 |
| | Massive(I) | 59 | 2,974 | 11,510 |
| Type | FewRel | 64 | 4,480 | 40,320 |
| | FewNerd | 58 | 3,789 | 50,000 |
| | FewEvent | 34 | 4,742 | 18,969 |
| Topic | StackEx | 121 | 4,156 | 50,000 |
| | ArxivS2S | 93 | 3,674 | 50,000 |
| | Reddit | 50 | 3,217 | 50,000 |
| Emotion | GoEmo | 27 | 5,940 | 23,485 |
| Domain | CLINC(D) | 10 | 4,500 | 15,000 |
| | MTOP(D) | 11 | 4,386 | 15,667 |
| | Massive(D) | 18 | 2,974 | 11,514 |

Table 1: Dataset statistics.

set up a max token of $10$. We then assign them to binary choices by directly checking whether one of the texts "Choice 1" or "Choice 2" is in the response. We also find that a very small amount of responses do not contain any choices and we discard them during fine-tuning. We use the Python API tool provided by OpenAI.

**Triplet Sampling.** For both small- or large-scale experiments, we set a budget of $Q = 1,024$ triplets. We set $\gamma_{\text{low}} = 20\%$ and $\gamma_{\text{high}} = 0$. For clustering methods, we fix hyperparameters of these algorithms across datasets in Stage 1. We choose agglomerative clustering with fixed distance threshold 67 for small-scale experiments on Instructor, and 77 on E5 (the embeddings are preprocessed by standard scaler). For large-scale datasets, we choose mini-batch K-means with fixed number of clusters 100 due to its lower latency. Clustering algorithms are implemented by scikit-learn (Pedregosa et al., 2011).

**Fine-tune Embedders.** In this work, we focus on two state-of-the-art pre-trained embedders: Instructor (Su et al., 2022) and E5 (Wang et al., 2022). We only use the large versions. Refer to Appendix D for details.

**Evaluation.** To reduce cost, we run CLUSTERLLM once for each dataset. We then run (mini-batch) K-means on (large) small-scale datasets for 5 seeds with ground truth K. We show two metrics. The first one is clustering accuracy calculated after Hungarian alignment (Kuhn, 1955) that permute prediction classes back to label classes. Another popular metric for clustering is normalized mutual information (NMI) that calculates mutual information between two assignments, and normalized by their individual entropies.

Table 2, part 1 — Intent Discovery and Emotion:

| Method | Bank77 ACC | Bank77 NMI | CLINC(I) ACC | CLINC(I) NMI | MTOP(I) ACC | MTOP(I) NMI | Massive(I) ACC | Massive(I) NMI | GoEmo ACC | GoEmo NMI |
|---|---|---|---|---|---|---|---|---|---|---|
| E5 (Wang et al., 2022) | 59.90(0.91) | 77.71(0.42) | 75.83(0.79) | 91.16(0.42) | 33.54(0.92) | 70.79(0.26) | 52.52(0.62) | 70.67(0.50) | 22.13(1.04) | 20.98(0.53) |
| SCCL-E (Zhang et al., 2021a) | 63.60(1.37) | 77.34(0.62) | 77.96(1.78) | 91.89(0.49) | 33.82(1.07) | 70.42(0.34) | 54.48(1.80) | 71.57(0.89) | 22.03(0.69) | 20.05(0.63) |
| self-supervise-E | 64.76(2.09) | 80.75(0.67) | 78.91(0.69) | 92.06(0.23) | 33.81(0.81) | 71.71(0.42) | 54.23(1.57) | 71.78(0.25) | 21.35(0.42) | 20.92(0.55) |
| CLUSTERLLM-E | 69.09(1.99) | 83.17(0.48) | 79.51(1.10) | 92.73(0.36) | 34.66(1.31) | 73.19(0.41) | 54.80(0.72) | 73.97(0.38) | 22.69(0.41) | 22.07(0.23) |
| CLUSTERLLM-E-iter | 70.13(1.34) | 84.16(0.36) | 80.48(0.93) | 92.92(0.29) | **37.22(1.18)** | **74.46(0.11)** | 56.08(1.01) | 74.39(0.21) | 22.22(1.15) | 22.23(0.17) |
| Instructor (Su et al., 2022) | 64.49(1.52) | 81.43(0.61) | 79.29(1.03) | 92.60(0.19) | 33.35(1.32) | 70.63(0.27) | 54.08(1.53) | 73.42(0.62) | 25.19(0.98) | 21.54(0.46) |
| SCCL-I (Zhang et al., 2021a) | 65.48(1.36) | 81.77(1.36) | 80.85(0.74) | 92.94(0.44) | 34.28(0.58) | 73.52(0.38) | 54.10(1.05) | 73.90(0.36) | **34.33(0.86)** | **30.54(0.68)** |
| self-supervise-I | 68.18(0.73) | 83.31(0.59) | 80.82(0.75) | 93.88(0.17) | 34.06(0.64) | 72.50(0.47) | 55.07(1.25) | 72.88(0.77) | 24.11(2.02) | 22.05(0.53) |
| CLUSTERLLM-I | 70.77(0.49) | 85.07(0.33) | 82.77(1.20) | 93.88(0.17) | 35.84(2.07) | 73.52(0.38) | 59.89(2.05) | 76.96(0.54) | 27.49(1.25) | 24.78(0.56) |
| CLUSTERLLM-I-iter | **71.20(1.59)** | **85.15(0.41)** | **83.80(0.41)** | **94.00(0.21)** | 35.04(0.97) | 73.83(0.79) | **60.69(0.96)** | **77.64(0.21)** | 26.75(1.76) | 23.89(0.68) |

Table 2, part 2 — Type Discovery and Topic Mining:

| Method | FewRel ACC | FewRel NMI | FewNerd ACC | FewNerd NMI | FewEvent ACC | FewEvent NMI | StackEx ACC | StackEx NMI | Reddit ACC | Reddit NMI |
|---|---|---|---|---|---|---|---|---|---|---|
| E5 (Wang et al., 2022) | 39.62(1.22) | 55.78(0.25) | 25.49(0.44) | 40.62(0.16) | 37.30(1.97) | 59.28(0.52) | 37.31(0.95) | 58.59(0.62) | 39.03(0.62) | 48.63(0.75) |
| SCCL-E (Zhang et al., 2021a) | 39.93(0.83) | 56.49(0.10) | 27.86(1.04) | 43.12(0.47) | 35.39(0.53) | 57.23(0.35) | 37.85(0.71) | 59.17(0.33) | 44.45(0.79) | 53.29(0.71) |
| self-supervise-E | 43.01(1.46) | 58.78(0.45) | 29.25(0.74) | 44.77(0.21) | 37.07(0.70) | 62.14(0.70) | 42.19(0.79) | 63.06(0.39) | 48.69(2.40) | 57.02(0.64) |
| CLUSTERLLM-E | 47.53(1.00) | 62.89(0.30) | 28.52(0.63) | 44.45(0.20) | 42.17(1.24) | 67.55(0.31) | 43.01(1.58) | 63.81(0.50) | 47.72(1.53) | 56.41(0.43) |
| CLUSTERLLM-E-iter | 52.95(1.15) | 67.64(0.29) | 32.16(0.83) | 48.16(0.25) | 44.64(0.90) | 70.74(0.35) | 43.93(0.93) | 64.66(0.36) | 49.05(1.15) | 57.35(0.31) |
| Instructor (Su et al., 2022) | 41.23(0.60) | 57.55(0.41) | 30.02(1.24) | 46.12(0.54) | 41.99(2.04) | 62.88(0.67) | 44.81(0.94) | 65.76(0.32) | 54.98(1.51) | 62.51(0.62) |
| SCCL-I (Zhang et al., 2021a) | 41.15(1.51) | 57.04(0.34) | 31.09(0.87) | 46.47(0.47) | 39.97(0.52) | 57.36(0.43) | 45.11(0.93) | 65.36(0.16) | 53.66(0.94) | 61.34(0.57) |
| self-supervise-I | 41.72(0.47) | 57.83(0.34) | 31.39(0.74) | 47.25(0.27) | 43.94(1.15) | 64.71(0.29) | 46.15(1.17) | 66.49(0.32) | 55.41(0.93) | 63.45(0.86) |
| CLUSTERLLM-I | 47.94(1.37) | 62.43(0.43) | 34.75(1.58) | 51.03(0.57) | 46.17(2.18) | 70.73(0.34) | 47.21(1.07) | 66.78(0.29) | 56.79(1.90) | 63.87(0.56) |
| CLUSTERLLM-I-iter | 51.22(1.43) | 65.53(0.40) | **40.60(0.77)** | **57.35(0.23)** | 50.60(0.79) | 73.89(0.47) | 47.75(1.24) | 67.08(0.30) | 57.02(0.87) | 63.92(0.38) |

Table 2, part 3 — Topic Mining, Domain Discovery, and Avg:

| Method | ArxivS2S ACC | ArxivS2S NMI | MTOP(D) ACC | MTOP(D) NMI | Massive(D) ACC | Massive(D) NMI | CLINC(D) ACC | CLINC(D) NMI | Avg ACC | Avg NMI |
|---|---|---|---|---|---|---|---|---|---|---|
| E5 (Wang et al., 2022) | 30.85(0.37) | 54.49(0.15) | 91.23(4.23) | 86.75(1.94) | **63.70(0.86)** | 66.20(0.72) | 59.64(2.73) | 57.23(0.66) | 47.72 | 61.35 |
| SCCL-E (Zhang et al., 2021a) | 32.78(0.69) | 55.77(0.30) | 92.28(0.18) | 86.01(0.22) | 61.22(1.15) | 65.27(0.99) | 59.88(2.74) | 56.21(0.94) | 48.82 | 61.70 |
| self-supervise-E | 34.41(0.52) | 57.09(0.26) | 91.40(4.61) | 88.20(1.71) | 61.79(2.50) | 65.02(1.32) | 57.29(2.51) | 57.07(0.88) | 49.87 | 63.60 |
| CLUSTERLLM-E | 34.93(0.36) | 57.90(0.09) | 89.18(5.25) | 86.19(2.05) | 60.73(2.81) | 66.15(0.69) | 56.25(2.94) | 56.32(0.81) | 50.77 | 64.77 |
| CLUSTERLLM-E-iter | **35.73(0.87)** | **58.42(0.38)** | 89.58(5.08) | 87.25(1.96) | 58.63(1.32) | 65.59(0.81) | **60.84(2.88)** | 58.55(2.09) | 52.40 | 66.18 |
| Instructor (Su et al., 2022) | 24.31(0.77) | 48.04(0.39) | 90.56(3.34) | 87.30(1.53) | 61.81(2.56) | 67.31(1.79) | 52.50(2.44) | 56.87(2.03) | 49.90 | 63.85 |
| SCCL-I (Zhang et al., 2021a) | 25.63(0.53) | 49.07(0.22) | 89.08(3.77) | 84.77(2.18) | 61.34(2.77) | 68.69(1.42) | 54.22(3.15) | 51.08(1.05) | 50.74 | 63.85 |
| self-supervise-I | 25.65(0.37) | 49.41(0.17) | 92.12(2.66) | 88.49(1.25) | 53.97(2.14) | **71.53(0.77)** | 58.58(2.56) | **60.84(1.04)** | 51.39 | 65.33 |
| CLUSTERLLM-I | 26.61(0.48) | 50.06(0.26) | **93.53(0.10)** | **89.36(0.11)** | 61.06(1.91) | 68.62(0.90) | 52.39(1.84) | 54.98(2.08) | 53.09 | 66.58 |
| CLUSTERLLM-I-iter | 26.34(0.38) | 50.45(0.19) | 92.13(3.81) | 89.23(1.21) | 60.85(4.33) | 68.67(1.59) | 51.82(1.91) | 54.81(1.15) | **53.99** | **67.53** |

Table 2: Comparison of clustering accuracy and NMI with known granularity for evaluation. Average over all 14 datasets are shown in the last two columns. Best results are bolded.

## 4.3 Compared Methods

**E5 and Instructor.** We directly apply (mini-batch) K-means on extracted embeddings from `instructor-large` and `e5-large`.

**self-supervise-I(E).** To verify that the performance improvement of CLUSTERLLM does not only come from domain-specific fine-tuning, instead of the more accurate triplet prediction. We propose a self-supervise fine-tuning that uses exactly the same triplets as CLUSTERLLM but only switch to self-supervised triplet predictions that select closest choices in embedding space.

**SCCL-I(E).** We also combine Instructor and E5 with SCCL (Zhang et al., 2021a), an unsupervised deep clustering algorithm that utilizes entire dataset for training. Notice that our method uses fewer data for training. See Appendix D for details.

## 4.4 Main Results

We show main results with small-scale datasets in Table 2. We show several variants of our method: CLUSTERLLM-I(E) adopt Instructor or E5 as embedders. CLUSTERLLM-I(E)-iter applies the entire framework twice in an iterative manner by using previous fine-tuned model as initialization and the 1,024 triplets inferred from new embeddings for fine-tuning. All of these use GPT-3.5 for prediction. We make the following observations: (1) CLUSTERLLM consistently improves upon both embedders. For example, CLUSTER-

| Type | Model | Bank77 | CLINC(I) | MTOP(I) | Massive(I) | GoEmo | FewRel | FewNerd |
|---|---|---|---|---|---|---|---|---|
| Random | #GT Triplets | 23 | 6 | 102 | 61 | 117 | 41 | 156 |
| | Instructor | 100 | 100 | 98.04 | 88.52 | 68.38 | 80.49 | 71.15 |
| | GPT3.5 | 100 | 100 | 85.29 | 85.25 | 68.38 | 85.37 | 82.05 |
| | Δ | (+0) | (+0) | (-12.75) | (-3.27) | (+0) | (+4.88) | (+10.90) |
| Entropy-based | #GT Triplets | 510 | 462 | 140 | 98 | 206 | 266 | 347 |
| | Instructor | 64.12 | 76.19 | 65.74 | 63.56 | 64.08 | 62.41 | 59.65 |
| | GPT3.5 † | 76.67 | 79.44 | 67.41 | 68.76 | 64.56 | 76.69 | 68.88 |
| | Δ | (+12.55) | (+3.25) | (+1.67) | (+5.20) | (+0.48) | (+14.28) | (+9.23) |
| | GPT4 | 79.41 | 80.74 | 76.04 | 74.84 | 61.65 | 87.22 | 82.13 |
| | Δ | (+15.29) | (+4.55) | (+10.30) | (+11.28) | (-2.43) | (+24.81) | (+22.48) |

| Type | Model | FewEvent | StackEx | Reddit | ArxivS2S | MTOP(D) | Massive(D) | CLINC(D) |
|---|---|---|---|---|---|---|---|---|
| Random | #GT Triplets | 105 | 14 | 40 | 22 | 184 | 148 | 189 |
| | Instructor | 98.10 | 85.71 | 80 | 95.45 | 96.74 | 80.41 | 76.72 |
| | GPT3.5 | 94.29 | 71.43 | 70 | 81.82 | 85.87 | 82.43 | 68.25 |
| | Δ | (-3.81) | (-14.28) | (-10) | (-13.63) | (-10.87) | (+2.02) | (-18.47) |
| Entropy-based | #GT Triplets | 259 | 271 | 92 | 145 | 144 | 108 | 208 |
| | Instructor | 70.66 | 68.27 | 61.98 | 59.31 | 70.31 | 63.82 | 75.06 |
| | GPT3.5 † | 83.78 | 71.22 | 63.28 | 73.79 | 69.79 | 72.09 | 75.78 |
| | Δ | (+13.12) | (+2.95) | (+1.30) | (+14.48) | (-0.52) | (+8.27) | (+0.72) |
| | GPT4 | 85.71 | 79.70 | 67.71 | 77.93 | 72.40 | 70.28 | 74.58 |
| | Δ | (+15.05) | (+11.43) | (+5.73) | (+18.62) | (+2.09) | (+6.46) | (-0.48) |

Table 3: Analysis on the triplet prediction accuracy († is used to produce results of CLUSTERLLM-I in Table 2). Red and green mean decreased or increased performances respectively. "#GT Triplets" means triplets that have ground truth (see Section 4.5 for details).

LLM-I increases the performance by 6.71% on FewRel. CLUSTERLLM-E increases the performance by 9.19 on Bank77. However, we do observe that on Massive(D) and CLINC(D), there are no improvements. (2) CLUSTERLLM outperforms deep clustering and self-supervise baselines. For instance, CLUSTERLLM-I surpasses self-supervise-I on most datasets except for two and it is also better than SCCL-I on 11 over 14 datasets. Furthermore, these improvements are consistent across both reported metrics. (3) Combined with the results in Appendix F, applying CLUSTERLLM iteratively is beneficial, emphasizing the potential of further improvements.

## 4.5 Analysis on Triplet Prediction Accuracy

We attribute the improvements on clustering quality to more accurate triplet predictions. In Table 3, we show the accuracy on predicted triplets that have ground truth (exactly one positive and one negative choices based on ground truth) with two different sampling methods. Random triplet sampling uniformly samples three random instances as query and two choices, and we guarantee the two choices are different from the anchor by filtering. Furthermore, we also show a selection accuracy with Euclidean distances between embeddings as a comparison. We observe that, GPT-3.5/4 consistently improves upon Instructor on high entropy ex-

amples, demonstrating our hypothesis. In contrast, with random sampling, the ground truth triplets is significantly fewer and the accuracy gap is much smaller or even decreases performance.

## 4.6 Ablation Study

**Clustering Quality.** We show ablation studies on CLUSTERLLM based on Instructor in Table 4. Specifically, we present results with 3 kinds of predictions on the same set of triplets for fine-tuning: GPT-3.5/4, replace triplet predictions of GPT-3.5 to ground truth on those triplets that have ground truth. We observe that GPT-4 marginally improves upon GPT-3.5 given the much higher cost. When provided with human labels, CLUSTERLLM-GT&GPT3.5 achieves the highest performance, which indicates the possibility for further improvement with more accurate predictions. We make similar observations for large-scale datasets in Table 6.

**Sampling Strategy.** In this section, we show ablation study on entropy-based sampling. In Figure 3, we observe that clustering accuracy increases when increasing entropies (or equally decreasing mean of interval) except for GoEmo. We make two hypothesis: (1) LLMs are much better than small embedders on harder instances. (2) high-entropy instances are generally more informative. In Table 4, we observe that training with randomly se-

| Method | Intent Discovery | | | | | | | | Emotion | |
|---|---|---|---|---|---|---|---|---|---|---|
| | Bank77 | | CLINC(I) | | MTOP(I) | | Massive(I) | | GoEmo | |
| | ACC | NMI | ACC | NMI | ACC | NMI | ACC | NMI | ACC | NMI |
| CLUSTERLLM-GPT3.5(random) | $59.88_{(2.56)}$ | $79.69_{(0.63)}$ | $74.40_{(0.91)}$ | $90.38_{(0.20)}$ | $28.05_{(1.69)}$ | $61.76_{(0.62)}$ | $51.66_{(2.41)}$ | $68.87_{(0.73)}$ | $28.62_{(1.95)}$ | $25.88_{(1.02)}$ |
| CLUSTERLLM-GPT3.5 | $70.77_{(0.49)}$ | $85.07_{(0.33)}$ | $82.77_{(1.20)}$ | $93.88_{(0.17)}$ | $35.84_{(2.07)}$ | $73.52_{(0.38)}$ | $\mathbf{59.89_{(2.05)}}$ | $76.96_{(0.54)}$ | $27.49_{(1.25)}$ | $24.78_{(0.56)}$ |
| CLUSTERLLM-GPT4 | $69.71_{(1.13)}$ | $84.68_{(0.40)}$ | $81.91_{(1.20)}$ | $93.76_{(0.24)}$ | $34.48_{(0.38)}$ | $73.57_{(0.40)}$ | $59.10_{(1.12)}$ | $76.59_{(0.41)}$ | $27.41_{(1.13)}$ | $23.77_{(0.42)}$ |
| CLUSTERLLM-GT&GPT3.5 | $\mathbf{71.35_{(1.97)}}$ | $\mathbf{85.12_{(0.45)}}$ | $\mathbf{84.00_{(1.04)}}$ | $\mathbf{94.34_{(0.30)}}$ | $\mathbf{36.86_{(0.42)}}$ | $\mathbf{75.36_{(0.08)}}$ | $59.27_{(1.43)}$ | $\mathbf{77.37_{(0.54)}}$ | $\mathbf{30.91_{(1.16)}}$ | $\mathbf{27.71_{(0.46)}}$ |

| Method | Type Discovery | | | | | | Topic Mining | | | |
|---|---|---|---|---|---|---|---|---|---|---|
| | FewRel | | FewNerd | | FewEvent | | StackEx | | Reddit | |
| | ACC | NMI | ACC | NMI | ACC | NMI | ACC | NMI | ACC | NMI |
| CLUSTERLLM-GPT3.5(random) | $40.65_{(0.89)}$ | $56.54_{(0.30)}$ | $27.15_{(0.53)}$ | $43.56_{(0.49)}$ | $44.23_{(1.72)}$ | $66.75_{(0.87)}$ | $40.81_{(0.94)}$ | $62.10_{(0.34)}$ | $54.60_{(2.23)}$ | $61.82_{(1.64)}$ |
| CLUSTERLLM-GPT3.5 | $47.94_{(1.37)}$ | $62.43_{(0.43)}$ | $34.75_{(1.58)}$ | $51.03_{(0.57)}$ | $46.17_{(2.18)}$ | $70.73_{(0.34)}$ | $\mathbf{47.21_{(1.07)}}$ | $66.78_{(0.29)}$ | $56.79_{(1.90)}$ | $63.87_{(0.56)}$ |
| CLUSTERLLM-GPT4 | $\mathbf{48.96_{(1.14)}}$ | $\mathbf{63.58_{(0.39)}}$ | $\mathbf{37.54_{(0.54)}}$ | $\mathbf{53.94_{(0.27)}}$ | $47.98_{(1.45)}$ | $71.32_{(0.70)}$ | $46.82_{(0.78)}$ | $66.72_{(0.11)}$ | $55.38_{(0.37)}$ | $63.45_{(0.49)}$ |
| CLUSTERLLM-GT&GPT3.5 | $48.91_{(1.20)}$ | $63.34_{(0.47)}$ | $37.27_{(0.61)}$ | $53.57_{(0.32)}$ | $\mathbf{48.12_{(1.52)}}$ | $\mathbf{72.31_{(0.84)}}$ | $47.55_{(1.17)}$ | $\mathbf{67.04_{(0.31)}}$ | $\mathbf{58.33_{(1.26)}}$ | $\mathbf{65.34_{(0.51)}}$ |

| Method | Topic Mining | | Domain Discovery | | | | | | Avg | |
|---|---|---|---|---|---|---|---|---|---|---|
| | ArxivS2S | | MTOP(D) | | Massive(D) | | CLINC(D) | | | |
| | ACC | NMI | ACC | NMI | ACC | NMI | ACC | NMI | ACC | NMI |
| CLUSTERLLM-GPT3.5(random) | $22.03_{(0.28)}$ | $45.50_{(0.16)}$ | $87.00_{(2.27)}$ | $82.09_{(1.54)}$ | $56.40_{(2.35)}$ | $64.39_{(1.12)}$ | $\mathbf{60.27_{(4.20)}}$ | $\mathbf{58.11_{(2.93)}}$ | 48.27 | 61.96 |
| CLUSTERLLM-GPT3.5 | $\mathbf{26.61_{(0.48)}}$ | $50.06_{(0.26)}$ | $\mathbf{93.53_{(0.10)}}$ | $\mathbf{89.36_{(0.11)}}$ | $61.06_{(1.91)}$ | $68.62_{(0.90)}$ | $52.39_{(1.84)}$ | $54.98_{(2.08)}$ | 53.09 | 66.58 |
| CLUSTERLLM-GPT4 | $26.16_{(0.22)}$ | $50.06_{(0.20)}$ | $92.04_{(2.67)}$ | $88.39_{(1.33)}$ | $60.16_{(2.97)}$ | $67.98_{(1.04)}$ | $57.45_{(2.48)}$ | $59.98_{(1.14)}$ | 53.22 | 66.98 |
| CLUSTERLLM-GT&GPT3.5 | $26.14_{(0.57)}$ | $\mathbf{50.19_{(0.33)}}$ | $92.26_{(3.62)}$ | $89.36_{(1.42)}$ | $\mathbf{61.65_{(3.50)}}$ | $\mathbf{69.51_{(1.50)}}$ | $52.87_{(2.63)}$ | $56.43_{(1.21)}$ | $\mathbf{53.96}$ | $\mathbf{67.64}$ |

Table 4: Ablation study on clustering quality with Instructor as backbone and known granularity for evaluation. See more results with large-scale datasets in Table 6.

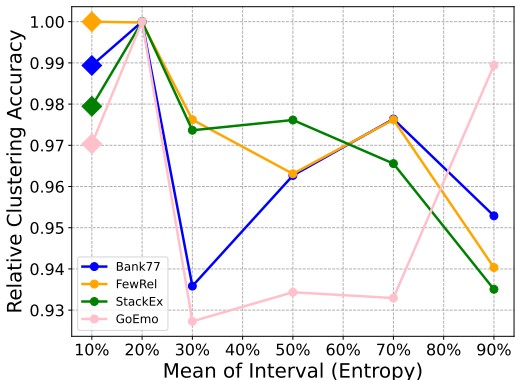

Figure 3: Relative clustering accuracy (divided by maximum for better aligning across datasets) of CLUSTER-LLM-GPT3.5 with different range of entropy selected. x-axis shows the mean of interval where interval length is set to 20%. For example, "mean of interval= 50%" means $\gamma_{high} = 40\%$ and $\gamma_{low} = 60\%$ (see Section 3.1.1). ♦ marks the setting for main experiments.

lected triplets even decreases performance, which demonstrates the cruciality of triplet sampling.

### 4.7 Determining Cluster Granularity

In this section, we show the results for determining cluster granularity. We evaluate on a subset of 8 datasets including various cluster granularities with $k_{max} = 200$ and $k_{min} = 2$. We compare with different methods that rely on clustering errors. For our methods, we show results with $\lambda = \{1, 3\}$ (except for GPT-4 to reduce costs), which involve

198 & 594 pairs in total respectively. To simulate experts for providing demonstrations, we directly sample 16 pairs from small-scale datasets when $\lambda = 3$ and then choose 2 positive and 2 negative as demonstrations. Notice that we use the same demonstrations for large-scale experiments. See more details in Appendix B.

We make several observations from Table 5 and Table 7: (1) Our methods have higher ranks. Most baseline methods predict similar number of clusters for domain and intent, while our methods can effectively distinguish between the two. For instance, on MTOP(I)/(D) in Table 5, BIC predicts number of clusters 69/64 while our method (GPT-3.5, $\lambda = 3$) predicts 92/18. (2) Increasing $\lambda$ generally helps (MTOP(D) in Table 5) but might not always make a large difference. (3) GPT-4 significantly improves upon GPT-3.5, probably due to its better understanding of demonstrations.

## 5 Related Works

**Clustering.** As a fundamental task in machine learning, clustering has been applied on diverse data types, including texts (Xu et al., 2015; Hadifar et al., 2019; Zhang et al., 2021a), images (Yaling Tao, 2021; Yang et al., 2016; Caron et al., 2018; Niu et al., 2020; Xie et al., 2016) and graphs (Huang et al., 2014; Chiang et al., 2019). Recent research has been shifted to deep cluster-

| Method | Bank77 | FewRel | Massive(I) | Massive(D) | MTOP(I) | MTOP(D) | CLINC(I) | CLINC(D) | Rank |
|---|---|---|---|---|---|---|---|---|---|
| GT #clusters | 77 | 64 | 59 | 18 | 102 | 11 | 150 | 10 | - |
| Silhouette (Rousseeuw, 1987) | 118 (53.25) | 10 (84.38) | 38 (35.59) | 41 (127.8) | 11 (89.22) | 11 (0) | 172 (14.67) | 163 (1530) | 10 |
| Elbow (Thorndike, 1953) | 53 (31.17) | 43 (32.81) | 45 (23.73) | 46 (155.6) | 33 (67.65) | 34 (209.1) | 66 (56.00) | 68 (580.0) | 9 |
| X-means (Pelleg et al., 2000) | 69 (10.39) | 30 (53.13) | 32 (45.76) | 34 (88.89) | 28 (72.55) | 27 (145.5) | 130 (13.33) | 135 (1250) | 8 |
| BIC (Goutte et al., 2001) | 123 (59.74) | 58 (9.38) | 56 (5.08) | 60 (233.3) | 69 (32.35) | 64 (481.8) | 167 (11.33) | 176 (1660) | 7 |
| ClusterSize (Zhang et al., 2021b) | 86 (11.69) | 71 (10.94) | 72 (22.03) | 90 (400.0) | 82 (19.61) | 85 (672.7) | 105 (30.00) | 106 (960.0) | 6 |
| Ours (GPT3.5,$\lambda = 1$) | 64 (16.88) | 46 (28.13) | 43 (27.12) | 90 (400.0) | 43 (57.84) | 40 (263.6) | 151 (0.67) | 96 (860.0) | 5 |
| Ours (GPT3.5,$\lambda = 3$) | 64 (16.88) | 46 (28.13) | 52 (11.86) | 37 (105.6) | 92 (9.80) | 18 (63.63) | 142 (5.33) | 107 (970.0) | 2 |
| Ours (GPT4,$\lambda = 1$) | 56 (27.27) | 46 (28.13) | 41 (30.51) | 20 (11.11) | 53 (48.04) | 8 (27.27) | 146 (2.67) | 39 (290.0) | 3 |
| Ours (GT,$\lambda = 1$) | 100 (29.87) | 91 (42.19) | 42 (28.81) | 18 (0) | 41 (59.80) | 11 (0) | 141 (6.00) | 39 (290.0) | 4 |
| Ours (GT,$\lambda = 3$) | 99 (28.57) | 94 (46.88) | 62 (5.08) | 20 (11.11) | 37 (63.73) | 11 (0) | 142 (5.33) | 31 (210.0) | 1 |

Table 5: Cluster granularity on small-scale datasets. Maximum & minimum number of clusters are set to 200 & 2. The results are shown in format of "[#clusters] (errors)". "Rank" column is computed with 1-level ranking (Colombo et al., 2022) with inverse errors. "GT" is ground truth. See results for large-scale datasets in Table 7.

ing (Zhou et al., 2022) which focuses on how to leverage deep neural network in clustering. Zhou et al. (2022) has categorized deep clustering research into four types including multi-stage (Yaling Tao, 2021; Huang et al., 2014), iterative (Yang et al., 2016; Caron et al., 2018; Van Gansbeke et al., 2020; Niu et al., 2022; Chang et al., 2017; Niu et al., 2020), generative (Dilokthanakul et al., 2016) and simultaneous (Xie et al., 2016; Zhang et al., 2021a; Hadifar et al., 2019) depended on how representation learning and clustering modules interact with each other. Most recently, a concurrent work (Wang et al., 2023) studies a similar problem by assigning instances to different explanations proposed by LLMs. Another recent work IDAS (Raedt et al., 2023) directly encode the concatenation of sentence and abstractive summarizations from LLMs for clustering.

**Clustering with Relations.** Clustering with relations has been explored in different situations. To start with, spectral clustering (Donath and Hoffman, 1972; Cheeger, 1970) makes use of similarity matrix where each entry measures the similarity between a pair of data. More recently, several works in deep clustering utilize relational supervision (Yang et al., 2016; Niu et al., 2020; Van Gansbeke et al., 2020; Chang et al., 2017) via pseudo-labelling which could be noisy. Another line of works that is closely related to ours is constrained clustering. It usually incorporates pairwise must-link or cannot-link constraints (Basu et al., 2004; Wagstaff et al., 2001; Basu et al., 2008; Zhang et al., 2020; Manduchi et al., 2021). Nonetheless, these constraints are often sampled from labels as a prior which significantly limits its application in our scenario. In this work, we study how to utilize contemporary API-based LLMs to infer relations.

**Pre-trained Embedding Model.** Generic pre-trained text embedding models (Reimers and Gurevych, 2019; Gao et al., 2021; Ni et al., 2022a,b) are widely applied in text similarity, classification, clustering and information retrieval. Recently, two embedding models E5 (Wang et al., 2022) and Instructor (Su et al., 2022) have shown superior performance on a popular benchmark (Muennighoff et al., 2022). Specifically E5 is pre-trained on web-scraped data pairs with contrastive objective. Instructor is pre-trained on 330 tasks with instructions. CLUSTERLLM aims at improving these models with LLMs.

## 6 Conclusion

In this paper, we study how to leverage API-based LLMs to guide small embedders for text clustering in order to benefit from high-level language capability of LLMs and user's instructions on clustering. We propose to prompt LLMs with two kinds of sentence relationship tasks: triplet task and pairwise task. Triplet task chooses the sentence that is most similar with anchor combining with a perspective instruction from users. The predicted triplets are used for fine-tuning small embedders. Pairwise task judges whether a pair of sentences belong to the same category hinted by few-shot demonstrations, and then the predictions are used to determine cluster granularity with a consistency measure. Extensive experiments show that our proposed framework CLUSTERLLM can improve clustering quality and propose reasonable cluster granularity at a negligible cost. However, CLUSTERLLM still relies on the embedding model itself, which is inefficient and inapplicable on black-box embedding models. We encourage future works to explore the potential of model-free training such as constrained clustering.

## Limitations

We list several limitations of our work that we hope to be improved in the future:

**Reliance on pre-trained embedder.** To find the most informative data, we have to rely on a pre-trained embedder that can indicate the largest clustering assignment entropy. We hope that self-supervise triplets and LLM-predicted triplets can be combined to solve such an issue.

**Computational cost for fine-tuning.** Our initial idea is to utilize constrained clustering which is a light-weight algorithm that do not need to update small embedders. However, the inevitable unstable training will be heavily affected by the errors in LLM predictions. We make a comprise by introducing embedder into fine-tuning to temporarily solve the issue, but we hope to reduce the computational cost in a future work.

**Sub-optimal performance on domain discovery.** We notice that on domain discovery datasets such as Massive(D) and CLINC(D), the performance is usually sub-optimal compared with original Instructor embedding. We provide discussions on this issue in Appendix H.

## Ethics Statement

Our work employs LLMs which are accessed through OpenAI APIs. For some applications, uploading privacy-sensitive data is risky and might require efforts to remove sensitive information.

## Acknowledgements

Our work is sponsored in part by NSF CAREER Award 2239440, NSF Proto-OKN Award 2333790, NIH Bridge2AI Center Program under award 1U54HG012510-01, Cisco-UCSD Sponsored Research Project, as well as generous gifts from Google, Adobe, and Teradata. Any opinions, findings, and conclusions or recommendations expressed herein are those of the authors and should not be interpreted as necessarily representing the views, either expressed or implied, of the U.S. Government. The U.S. Government is authorized to reproduce and distribute reprints for government purposes not withstanding any copyright annotation hereon.

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

## A Details of Scaling up Hierarchical Clustering

A major drawback of hierarchical clustering is its $\mathcal{O}(N^3)$ time complexity which makes the algorithm hard to be deployed on larger datasets. However, since we are only interested in a specific range of granularity in our scenario, the hierarchical clustering can start from an intermediate step. We address this issue by first running mini-batch K-means with $k_{\max}$ and then run hierarchical clustering with current assignments as inputs. Specifically, we use agglomerative clustering with ward's method (Ward Jr, 1963). We first calculate distances between each pair of clusters according to Murtagh and Contreras, 2011 and then provide them as inputs to *nearest neighbor chain* algorithm. Finally the returned hierarchy is combined with the K-means assignments to infer clusters.

## B More Details about Determining Cluster Granularity

Previous methods often employ clustering errors as a metric and they ignore user's need on the granularity. **Silhouette coefficient** (Rousseeuw, 1987) indicates the clustering quality without ground truths, which exploits the inter-cluster distance with nearest clusters and the intra-cluster distance. We find the granularity by choosing the one with the best silhouette coefficient. **Elbow method** (Thorndike, 1953) is a heuristic method that plots the clustering error with respect to different levels of granularity in the hierarchy. And then the best granularity is determined with the largest elbow length. **X-means** (Pelleg et al., 2000) is a variation of K-means that starts with the lowest number of clusters, and then repeatedly attempt to split the clusters by running 2-means on them and evaluate with Bayesian Information Criterion (BIC) (Goutte et al., 2001). **BIC** (Goutte et al., 2001) calculates BIC for each of the granularity. **Cluster-Size** (Zhang et al., 2021b) uses a confidence threshold to filter small clusters starting from the maximum number of cluster. For all methods, we use the same fine-tuned embeddings (CLUSTERLLM-I in Table 2). The same cluster hierarchy is used (except for X-means that relies on K-means), which is either acquired from hierarchical clustering for small-scale or our proposed two-step method in Section 3.2 for large-scale. For out methods, the weight in F-beta score is set to 0.92 through empirical selection on Bank77. Because of the high

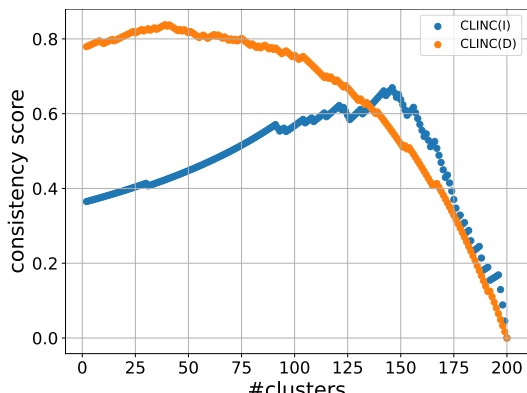

Figure 4: Consistency score v.s. various number of clusters with GPT-4 and $\lambda = 1$.

latency, results for Silhouette and X-means are not shown on large-scale datasets. After sampling 16 data pairs, we tend to choose positives with finer granularity or negatives with coarser granularity. We also consider the sentence length to minimize the cost. We use label names as justifications and we always put 2 positive before 2 negative (See Table 12 bottom).

## C Analysis for Determining Granularity

**Prompt Design.** We show the analysis results of prompt design for determining granularity in Table 8. We experiment with two settings: (1) remove justification for all demonstrations and only keep the "Yes" or "No" answer. (2) remove all demonstrations and any granularity-related words (such as domain)[3]. We observe that demonstrations are necessary and adding justifications have a positive impact.

**Visualization of Consistency Score.** We visualize consistency score with respect to the number of clusters. The consistency scores exhibit continuous variations and peak at the best number of clusters.

## D Details of Embedders and Fine-tuning

For all the experiments (including those with or without fine-tuning), we use large version of both Instructor and E5 (i.e. `hkunlp/instructor-large` & `intfloat/e5-large`). For Instructor, we use the same or similar prompt as original paper. See Table 10.

For fine-tuning, we adopt the same hyper-parameters as in (Su et al., 2022), but modify the

---

[3]After removing, the prompt will be in the format "Determine whether the two sentences below belong to the same category."

learning rate to $2e-6$, the maximum gradient steps to $3,840$ for Instructor ($\sim 15$ epochs) and $1,280$ for E5, and batch size to $4$. We choose this gradient in the begining of our experiments by observing no performance increase after that on several datasets. Training is conducted with a single NVIDIA Quadro RTX 8000 GPU.

For SCCL-I(E), we change the maximum token length to 128 due to the limited compute resource[4]. We use the same learning rate $2e-6$ as before for Instructor and $2e-7$ for E5 since we found that the performance is unstable with large learning rate. Batch size is set to 16 and we evaluate representations with K-means after 200 iterations. Also notice that we do not interrupt prompts in Instructor during data augmentation.

## E Description of Datasets

**Bank77** (Casanueva et al., 2020) is a popular dataset in intent discovery that focuses on creating fine-grained intent categories for a single-domain, "banking". **CLINC(I)** (Larson et al., 2019) is originally created for detecting utterances that falls outside of supported intents. The dataset also contains multiple domains, such as "travel", "utility" and "work". In this experiment, we discard all the out-of-scope utterances and only focus on in-domain ones. Moreover, we create a domain discovery dataset **CLINC(D)** that uses domains as labels. **Massive(I)/(D)** (FitzGerald et al., 2022) and **MTOP(I)/(D)** (Li et al., 2021) are both from MTEB (Muennighoff et al., 2022). Here "I" denotes intent and "D" for domain (or scenario). These datasets are originally used for classification but are adapted here for clustering. We also remove those intents with only a few instances and keep English-only data. For all datasets, we use the train & test sets as large- & small- scale datasets respectively. For **FewRel** (Gao et al., 2019) and **FewEvent** (Deng et al., 2020), we first randomly split datasets into train & test sets, and then sample from train set as large-scale and test set as small-scale. For **FewNerd** (Ding et al., 2021), we use the original train & test splits. For **StackEx**, **Reddit** (Geigle et al., 2021) and **ArxivS2S**, we combine all the splits into a single dataset and remove topics that only have few instances. Finally, the datasets are randomly splitted into large- & small- scale versions. To show the dataset balancy, we show the

[4]We also tried maximum token length 512 with smaller batch size, but the performance is not better.

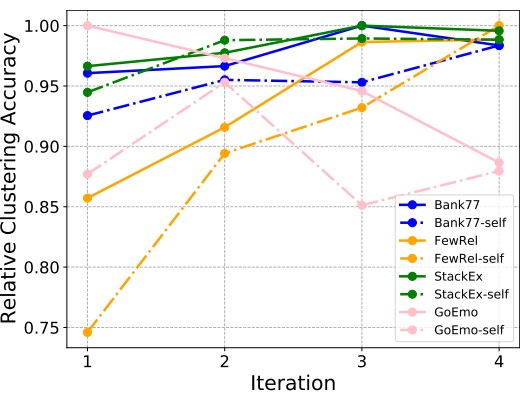

Figure 5: Relative clustering accuracy (divided by maximum for better aligning across datasets) of CLUSTER-LLM-GPT3.5 over 4 iterations.

entropy of class distribution in Table 9.

## F Results of More Iterations

We show the results over $4$ iterations of CLUS-TERLLM in Figure 5. During iteration, we sample triplets from previously fine-tuned embedding space and continue to fine-tune the model with previous checkpoint as initialization. We also show the self-supervise results using the same checkpoint fine-tuned with GPT-3.5 predictions as initialization at each iteration. We observe that using GPT-3.5 predictions is almost always beneficial. The performance generally increases and saturate at the fourth iteration with the exception of GoEmo.

## G More Related Works

**Generalized Category Discovery (GCD).** GCD (Vaze et al., 2022; Lin et al., 2020; Zhang et al., 2021b, 2022; Mou et al., 2022; An et al., 2022) assume partial known classes with annotations which can also be used to infer user's requirement on clustering. As an infant research area, most previous works employ pseudo-labelling, via optimal transport (Rizve et al., 2022b; Yang et al., 2022), similarity learning (Rizve et al., 2022a; Cao et al., 2022) or prototype-based learning (Sun and Li, 2022). Furthermore, new intent discovery (Zhang et al., 2022, 2021b, 2023; An et al., 2022; Lin et al., 2020) is proposed to study a similar research problem in the domain of intent detection. Most recently, Hogan et al., 2023 adapts the setting into relation type discovery. However, GCD relies on sufficient annotated and unlabeled data for training. In contrast, CLUSTERLLM seeks for minimal supervision and studies a setting with controlled

| Method | Intent Discovery | | | | | | | | Emotion | |
|---|---|---|---|---|---|---|---|---|---|---|
| | Bank77 | | CLINC(I) | | MTOP(I) | | Massive(I) | | GoEmo | |
| | ACC | NMI | ACC | NMI | ACC | NMI | ACC | NMI | ACC | NMI |
| Instructor | 60.30(2.39) | 78.37(0.78) | 79.52(1.96) | 92.65(0.42) | 35.53(1.05) | 70.78(0.74) | 54.72(2.00) | 72.29(0.61) | 24.02(1.11) | 20.15(0.19) |
| +self-supervise | 61.48(2.84) | 79.28(1.02) | 81.87(0.97) | 93.55(0.41) | 35.27(1.53) | 71.88(0.66) | 58.30(1.42) | 73.73(0.56) | 24.34(1.25) | 21.17(0.56) |
| +CLUSTERLLM-GPT3.5 | 65.47(2.28) | 81.60(0.92) | 82.29(1.09) | 93.91(0.11) | 36.80(0.83) | 73.16(0.37) | 57.70(2.92) | 74.24(0.68) | 25.23(1.21) | 22.19(0.42) |
| +CLUSTERLLM-GT&GPT3.5 | 67.30(1.35) | 82.46(0.17) | 80.90(1.58) | 93.80(0.25) | 37.75(1.64) | 74.51(0.51) | 58.92(1.80) | 75.07(0.43) | 27.96(2.59) | 25.90(0.69) |

| Method | Type Discovery | | | | | | Topic Mining | | | |
|---|---|---|---|---|---|---|---|---|---|---|
| | FewRel | | FewNerd | | FewEvent | | StackEx | | Reddit | |
| | ACC | NMI | ACC | NMI | ACC | NMI | ACC | NMI | ACC | NMI |
| Instructor | 41.38(0.75) | 53.97(0.23) | 29.62(1.02) | 42.83(0.31) | 41.42(2.09) | 61.13(1.11) | 46.76(0.80) | 61.20(0.37) | 55.04(2.69) | 59.55(0.98) |
| +self-supervise | 41.09(0.99) | 54.02(0.29) | 30.57(0.18) | 43.64(0.26) | 45.54(1.70) | 64.70(0.50) | 46.24(0.46) | 60.78(0.26) | 56.45(1.59) | 60.11(0.39) |
| +CLUSTERLLM-GPT3.5 | 47.22(0.89) | 59.87(0.21) | 33.86(1.19) | 46.91(0.52) | 47.55(1.51) | 70.21(0.59) | 47.42(1.35) | 61.34(0.30) | 55.47(2.44) | 58.73(1.09) |
| +CLUSTERLLM-GT&GPT3.5 | 49.20(0.47) | 61.20(0.30) | 34.85(1.42) | 48.59(0.33) | 46.90(1.77) | 71.51(0.83) | 48.12(0.93) | 62.04(0.36) | 57.95(1.96) | 61.39(0.92) |

| Method | Topic Mining | | Domain Discovery | | | | | | Avg | |
|---|---|---|---|---|---|---|---|---|---|---|
| | ArxivS2S | | MTOP(D) | | Massive(D) | | CLINC(D) | | | |
| | ACC | NMI | ACC | NMI | ACC | NMI | ACC | NMI | ACC | NMI |
| Instructor | 24.55(0.42) | 39.86(0.06) | 85.01(2.18) | 83.96(0.93) | 56.11(5.07) | 61.86(3.27) | 51.74(2.97) | 55.28(2.56) | 48.98 | 60.99 |
| +self-supervise | 24.49(0.75) | 40.70(0.16) | 89.54(4.56) | 86.69(2.09) | 58.14(3.88) | 64.49(2.28) | 50.93(4.11) | 53.01(4.48) | 50.30 | 61.98 |
| +CLUSTERLLM-GPT3.5 | 25.60(0.51) | 41.50(0.14) | 84.08(3.34) | 84.57(1.97) | 58.14(3.97) | 65.50(1.35) | 50.12(4.13) | 53.46(2.13) | 51.21 | 63.37 |
| +CLUSTERLLM-GT&GPT3.5 | 25.29(0.41) | 41.81(0.13) | 84.99(4.24) | 86.59(1.53) | 58.55(1.40) | 65.66(1.00) | 57.08(1.75) | 58.35(1.78) | 52.55 | 64.92 |

Table 6: Ablation study on clustering quality for large-scale datasets.

| Method | Bank77 | FewRel | Massive(I) | Massive(D) | MTOP(I) | MTOP(D) | CLINC(I) | CLINC(D) | Rank |
|---|---|---|---|---|---|---|---|---|---|
| GT #clusters | 77 | 64 | 59 | 18 | 102 | 11 | 150 | 10 | - |
| Elbow (Thorndike, 1953) | 50 (35.06) | 38 (40.63) | 47 (20.34) | 45 (150.0) | 35 (65.69) | 35 (218.2) | 64 (57.33) | 71 (610.0) | 7 |
| BIC (Goutte et al., 2001) | 183 (137.7) | 183 (185.9) | 148 (150.8) | 141 (683.3) | 169 (65.69) | 170 (1445) | 179 (19.33) | 178 (1680) | 8 |
| ClusterSize (Zhang et al., 2021b) | 90 (16.88) | 95 (23.38) | 79 (33.90) | 81 (350.0) | 84 (17.65) | 83 (654.5) | 105 (30.00) | 109 (990.0) | 5 |
| Ours (GPT3.5,$\lambda = 1$) | 79 (2.60) | 32 (50.00) | 122 (106.8) | 55 (205.6) | 56 (50.00) | 13 (18.18) | 143 (4.67) | 118 (1080) | 6 |
| Ours (GPT3.5,$\lambda = 3$) | 79 (2.60) | 52 (18.75) | 108 (83.05) | 54 (200.0) | 61 (40.20) | 20 (81.81) | 145 (3.33) | 118 (1080) | 4 |
| Ours (GPT4,$\lambda = 1$) | 64 (16.88) | 40 (37.50) | 58 (1.69) | 16 (11.11) | 24 (76.47) | 8 (27.27) | 143 (4.67) | 32 (220.0) | 3 |
| Ours (GT,$\lambda = 1$) | 59 (23.38) | 79 (23.44) | 61 (3.39) | 17 (5.56) | 27 (73.53) | 11 (0) | 148 (1.33) | 32 (220.0) | 2 |
| Ours (GT,$\lambda = 3$) | 77 (0) | 78 (21.88) | 54 (8.47) | 37 (105.6) | 19 (81.37) | 11 (0) | 148 (1.33) | 32 (220.0) | 1 |

Table 7: Inferred granularity on large-scale datasets. The setting is the same as in Table 5.

| Method | small-scale | | | | large-scale | | | |
|---|---|---|---|---|---|---|---|---|
| | Massive(I) | Massive(D) | CLINC(I) | CLINC(D) | Massive(I) | Massive(D) | CLINC(I) | CLINC(D) |
| GT | 59 | 18 | 150 | 10 | 59 | 18 | 150 | 10 |
| Ours | 41 | 20 | 146 | 39 | 58 | 16 | 143 | 32 |
| w/o Justification | 41 | 50 | 137 | 41 | 80 | 36 | 129 | 32 |
| w/o Demonstration | 41 | 64 | 141 | 108 | 77 | 74 | 120 | 105 |

Table 8: Prompt designs in determining granularity. We use the Instructor embedding with prompts, and we report results of GPT-4 with $\lambda = 1$.

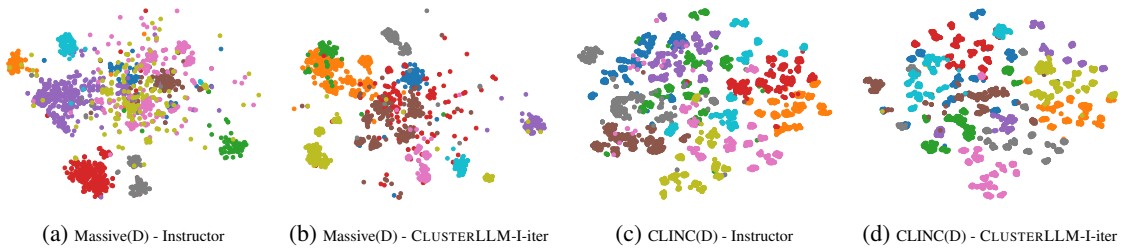

(a) Massive(D) - Instructor  (b) Massive(D) - CLUSTERLLM-I-iter  (c) CLINC(D) - Instructor  (d) CLINC(D) - CLUSTERLLM-I-iter

Figure 6: Scatter plots for t-SNE of embeddings. We select 10 classes from each datasets, denoted by colors.

| Task | Name | entropy(small) | entropy(large) |
|------|------|----------------|----------------|
| Intent | Bank77 | 1.00 | 1.00 |
|        | CLINC(I) | 1.00 | 1.00 |
|        | MTOP(I) | 0.74 | 0.75 |
|        | Massive(I) | 0.91 | 0.92 |
| Type | FewRel | 1.00 | 1.00 |
|      | FewNerd | 0.82 | 0.82 |
|      | FewEvent | 0.85 | 0.85 |
| Topic | StackEx | 0.98 | 0.98 |
|       | ArxivS2S | 1.00 | 1.00 |
|       | Reddit | 1.00 | 1.00 |
| Emotion | GoEmo | 0.91 | 0.91 |
| Domain | CLINC(D) | 1.00 | 1.00 |
|        | MTOP(D) | 0.98 | 0.98 |
|        | Massive(D) | 0.94 | 0.94 |

Table 9: Entropy of class distribution.

| Dataset | Prompt |
|---------|--------|
| Bank77 | Represent the bank purpose for retrieval: |
| CLINC(I) | Represent the sentence for retrieving the purpose: |
| FewRel | Represent the relation between two entities for retrieval: |
| FewNerd | Represent the entity type for retrieval: |
| FewEvent | Represent the event type for retrieval: |
| StackEx | Represent the question for retrieval: |
| ArxivS2S | Represent the science statement for retrieval: |
| GoEmo | Represent an emotion sentence for retrieval: |
| Massive(I) | Represent the sentence for retrieving the purpose: |
| MTOP(I) | Represent the sentence for retrieval: |
| Reddit | represent a reddit community title: |
| Massive(D) | Represent the scene for retrieval: |
| MTOP(D) | Represent a sentence: |
| CLINC(D) | Represent a sentence: |

Table 10: Prompts for Instructor.

computation- & data- cost.

**LLMs as Annotators.** Recent instruction-tuned LLMs, such as ChatGPT, have been shown to have the ability to reproduce or improve human-generated labels (Gilardi et al., 2023; He et al., 2023; Zhu et al., 2023). Furthermore, several works dedicate to fine-tune models with feedbacks from LLMs (Cheng et al., 2023; Bai et al., 2022). This paper instead focuses on clustering tasks.

## H  Sub-optimal Performance on Domain Discovery

We noticed that the performance of domain discovery (MTOP(D), Massive(D) and CLINC(D)) is barely improved or even decreased with CLUSTERLLM from original embedders (see Table 2). Furthermore, the ablation studies reveal that even with CLUSTERLLM-GT&GPT3.5, clustering performance is not as good as self-supervise or CLUSTERLLM-random (see CLINC(D) in Table 4 and MTOP(D) in Table 6). We also observe that, CLUSTERLLM-I-iter will further decrease the performance (see Massive(D) in Table 2). While we do not have rigorous explanations, one hypothesize is that the embedding space after fine-tuning

tends to be more compact than before and forming small cliques, making it better for clustering fine-grained clusters but not for coarse-grained clusters. We showcase scatterplots on two datasets with both Instructor and CLUSTERLLM-I-iter. It can be observed that the clusters in embedding space are tighter and more separated especially on CLINC(D).

## I  Dataset Leakage

Since LLMs like ChatGPT are trained on web-scraped texts from internet, it is likely they already have access to our evaluation datasets during training. For example, the topic mining datasets use StackExchange, Reddit and Arxiv tags as labels which is freely available online. However, as observed in Table 3, the performance of triplet prediction on these datasets are often far from perfect. Furthermore, the other datasets like Bank77 are synthesized datasets which is not accessible during training. FewRel is collected from Wikipedia corpus but their labels are not easily accessible. Similarly, GoEmo is collected from Reddit but the emotion labels are not accessible during training. Thus, dataset leakage is not a primary concern of this paper.

| Dataset | Prompt |
|---|---|
| Bank77 | Select the banking customer utterance that better corresponds with the Query in terms of intent. |
| CLINC(I) | Select the customer utterance that better corresponds with the Query in terms of intent. |
| FewRel | Select the example that better corresponds with the Query in terms of relation type. |
| FewNerd | Select the example that better corresponds with the Query in terms of entity type. |
| FewEvent | Select the example that better corresponds with the Query in terms of event type. |
| StackEx | Select the StackExchange question that better corresponds with the Query in terms of topic. |
| ArxivS2S | Select the Arxiv paper title that better corresponds with the Query in terms of domain. |
| GoEmo | Select the sentence that better corresponds with the Query in terms of emotion expressed. |
| Massive(I) | Select the user utterance that better corresponds with the Query in terms of intent. |
| MTOP(I) | Select the user utterance that better corresponds with the Query in terms of intent |
| Reddit | Select the Reddit question that better corresponds with the Query in terms of topic. |
| Massive(D) | Select the user utterance that better corresponds with the Query in terms of scenario. |
| MTOP(D) | Select the user utterance that better corresponds with the Query in terms of domain. |
| CLINC(D) | Select the customer utterance that better corresponds with the Query in terms of domain. |

Table 11: Prefix of prompts for triplet task. Notice while we use different prompts for domain and intent (such as CLINC(I) and CLINC(D)) in our experiments, they might be used interchangeably.

| Dataset | Prompt |
|---|---|
| Triplet Task | Select the banking customer utterance that better corresponds with the Query in terms of intent.

Query: Should i reinstall the payment app?
Choice 1: I've received my card so now I need to know how to sync it to the app.
Choice 2: Can I still use the app if I switched phones?

Please respond with 'Choice 1' or 'Choice 2' without explanation. |
| Pairwise Task | [Example1]
Sentence 1: I would like to see the source of my money.
Sentence 2: My source of funds need verified.
Yes. Because both intents are verify source of funds.

[Example2]
Sentence 1: Is there a fee for topping up
Sentence 2: What are the top up charges for US cards?
Yes. Because both intents are top up by card charge.

[Example3]
Sentence 1: Can I reactivate my lost card that I found this morning in my jacket pocket?
Sentence 2: how to activate card?
No. Because Sentence 1 has intent card linking and Sentence 2 has intent activate my card.

[Example4]
Sentence 1: What will I be charged for a physical card?
Sentence 2: My card is about to expire and I need to know how much it costs and how long ...
No. Because Sentence 1 has intent order physical card and Sentence 2 has intent card ...

Determine whether the intents of two banking customer utterances
below belong to the same intent category using above examples.

Sentence 1: $1 extra has been charged on my statement, why is that?
Sentence 2: Will it automatically top-up if there isn't much money left?

Please respond with 'Yes' or 'No' without explanation. |

Table 12: One example from Bank77 on both triplet task and pairwise task.