# OpenReview forum: "ClusterLLM: Large Language Models as a Guide for Text Clustering"
_EMNLP/2023/Conference — EMNLP 2023 Main_

### Official Review · Reviewer_odh6 · 2023-07-22

**Soundness:** 4

**Excitement:**

4: Strong: This paper deepens the understanding of some phenomenon or lowers the barriers to an existing research direction.

**Paper Topic And Main Contributions:**

The paper introduces CLUSTERLLM, a text clustering framework that utilizes feedback from an instruction-tuned LLM. ClusterLLM  leverages the emergent capabilities of ChatGPT even when its embeddings are inaccessible. The authors prompt ChatGPT to gain insights into clustering perspectives using hard triplet questions and to help with clustering granularity using pairwise questions.

**Questions For The Authors:**

-In your results tables you state that some results of the compared models are done with BERT instead of the pre-trained embedding models used for the presented technique. If that is the case, isn't it quite obvious that any comparison between K-Means/ClusterLLM using E5/instructor and the GCD methods  using BERT are unfair? How do the GCD methods perform when using the same embedding model as ClusterLLM?

I am unsure about the advantages and differences between the presented approach and any few-shot model trained with labeled documents from ChatGPT. Consequently, the benefits of using the triplet technique are not entirely clear to me. Could you please clarify or do you maybe have already performed experiments on that regard?

**Reasons To Accept:**

- Overal sound idea
- Good experimental results

**Reasons To Reject:**

- Poorly written, with many grammatical errors and illogical sentences.
- Not sure how a few-shot method (e.g. SetFit) would compare with documents "labeled" from GPT without the triplet technique. Benefit of the method is not quite clear to me.
- The fairness of the experimental comparison and chosen models is unclear.

**Reproducibility:**

3: Could reproduce the results with some difficulty. The settings of parameters are underspecified or subjectively determined; the training/evaluation data are not widely available.

**Reviewer Confidence:**

2: Willing to defend my evaluation, but it is fairly likely that I missed some details, didn't understand some central points, or can't be sure about the novelty of the work.

**Typos Grammar Style And Presentation Improvements:**

Line 205 -> hypothesis instead of hypothesize
Line 210 sample two data ...?

... a lot more grammatically missing words in sentences, e.g. lines 73, 388, 412 and many many more...

---

> ### Author Rebuttal · Authors · 2023-08-27
>
> We thank you for your comments and feedback. We address your concerns here.
>
> > Poorly written, with many grammatical errors and illogical sentences.
>
> Answer: As also pointed out by Reviewer s3g2 and Reviewer vYLX, the paper presentation needs to be significantly improved in order to increase readability. The authors want to apologize for the inclarity. And given one more page, we would like to improve especially on the following aspects: (1) in order to present holistic results across datasets, the splitted tables will either appear on the same page or merged together. We will also avoid discussing Appendix in the main paper; (2) both clustering accuracy and NMI will be presented for all datasets and models in the main results; (3) the paper will be rearranged in a better fashion to include the most important sections into the main paper; (4) the authors will meticulously go over the paper and correct grammatical errors.
>
> > Not sure how a few-shot method (e.g. SetFit) would compare with documents "labeled" from GPT without the triplet technique. Benefit of the method is not quite clear to me.
>
> Answer: The results on few-shot methods do not serve as a direct comparison with ClusterLLM since our method does not have access to annotated data. As stated in Line 389-393, we wish to show that a better pre-trained encoder is worth hundreds of annotated data which may not be widely known in the area of especially intent discovery. However, it is a good point that perhaps such a comparison is not the focus of this paper, and therefore should better be moved to Appendix. Additionally, we will include the results of SetFit and different embedding models in the next version.
>
> > The fairness of the experimental comparison and chosen models is unclear.
>
> Answer: The main baselines of this paper are the self-supervised method and SCCL (Line 398-410). We would like to show that with carefully sampled triplets, we can improve clustering performance with a limited amount of LLM predictions. The self-supervised method uses the same triplets as ours but annotated by the embedding itself. SCCL is a well-known clustering method relying on contrastive learning. It is interesting to see that with only 1,024 triplets predicted by LLMs, it can already significantly improve clustering performance while also outperforming these baselines on most datasets (see Table.2 and discussions in Section 4.4).
>
> > -In your results tables you state that some results of the compared models are done with BERT instead of the pre-trained embedding models used for the presented technique. If that is the case, isn't it quite obvious that any comparison between K-Means/ClusterLLM using E5/instructor and the GCD methods using BERT are unfair? How do the GCD methods perform when using the same embedding model as ClusterLLM?
> >
> > I am unsure about the advantages and differences between the presented approach and any few-shot model trained with labeled documents from ChatGPT. Consequently, the benefits of using the triplet technique are not entirely clear to me. Could you please clarify or do you maybe have already performed experiments on that regard?
>
> Answer: As stated above, our aim is not to compare with few-shot models, but rather to show the importance of pre-training. However, we are willing to show this result in the Appendix and leave space for more important results.
>
> > Line 205 -> hypothesis instead of hypothesize Line 210 sample two data ...?
> >
> > ... a lot more grammatically missing words in sentences, e.g. lines 73, 388, 412 and many many more…
>
> Answer: As stated above, we will meticulously go over the paper and correct these errors.

---

### Official Review · Reviewer_vYLX · 2023-07-25

**Soundness:** 4

**Excitement:**

3: Ambivalent: It has merits (e.g., it reports state-of-the-art results, the idea is nice), but there are key weaknesses (e.g., it describes incremental work), and it can significantly benefit from another round of revision. However, I won't object to accepting it if my co-reviewers champion it.

**Paper Topic And Main Contributions:**

The paper presents a novel algorithm ClusterLLM which leverages LLMs for improving text clustering. Given a text embedder, ClusterLLM prompts LLM for correct assignment of points to clusters and fine tunes the embedder based on this information. It considers 2 broad tasks for it : triplet querying which aims to capture the clustering semantics/perspective and pairwise query which deals with the granularity within a cluster.

**Questions For The Authors:**

1. Did you try any other metric for sampling triplets apart from random or entropy-based? The observation the “random sampling inputs for LLM annotations is not optimal/might even hurt performance” seems to be common across multiple works.

**Reasons To Accept:**

1. The paper advances  the current clustering literature by leveraging the recent developments in instruction-tuned LLMs.

2. Interestingly, they show how naive random sampling for triplet queries might not lead to better clustering performance and how entropy based triplet selection can help alleviate this issue.

3. Empirically the paper shows gains over current SOTA across multiple datasets and includes a thorough analysis.

**Reasons To Reject:**

1. The cluster granularity aspect of the algorithm seems under-explored/motivated and I couldn’t understand it well enough. Probably the paper would be better written as solely a clustering paper inline with prior work and exploration on granularity could come in appendix or as a potential future work.

2. Grammatically the paper can be improved a lot. Hope the authors will read thoroughly and rectify these mistakes.

**Reproducibility:**

4: Could mostly reproduce the results, but there may be some variation because of sample variance or minor variations in their interpretation of the protocol or method.

**Reviewer Confidence:**

3: Pretty sure, but there's a chance I missed something. Although I have a good feel for this area in general, I did not carefully check the paper's details, e.g., the math, experimental design, or novelty.

**Typos Grammar Style And Presentation Improvements:**

Spliting main result tables into multiple tables creates a bit of confusion for the reader. I found myself lost in table numbers and flipping pages. Also would be good to mark Supp tables as "Supp. Table 5" to make it explicit.

In general i feel the authors can do a better job of presenting the results in a more streamlined fashion with clear agenda contained within the page limits of the paper which wouldn't require the reader to necessarily refer the appendix.

---

> ### Author Rebuttal · Authors · 2023-08-27
>
> We thank you for your comments and feedback. We are glad to see that you notice the advancements we wish to provide on clustering literature. We address your concerns here.
>
> > The cluster granularity aspect of the algorithm seems under-explored/motivated and I couldn’t understand it well enough. Probably the paper would be better written as solely a clustering paper inline with prior work and exploration on granularity could come in appendix or as a potential future work.
>
> Answer: The motivation is that predicted pairwise relations from the LLM can be used to represent granularity of clustering. A set of pairs are first sampled from hierarchical clustering, and then predicted by LLM with binary “Yes/No” answers indicating whether a pair of sentences belong to the same cluster. We then compare these pairwise predictions against each level of granularity in hierarchical clustering to acquire the granularity with the highest consistency (F-beta score in the paper). To be more specific, each granularity is itself a clustering of the dataset. To calculate consistency, we count the number of predicted pairs that have the same assignments as current clustering of the granularity. The authors hope to convey a simple message that at least there exists a way to automatically estimate granularity with a proper prompt. As stated in the response to Reviewer s3g2, we would like to include more analysis in this work and we also encourage more future works that can bring more insights.
>
> > Grammatically the paper can be improved a lot. Hope the authors will read thoroughly and rectify these mistakes.
>
> Answer: As also pointed out by Reviewer s3g2 and Reviewer odh6, the paper presentation needs to be significantly improved in order to increase readability. The authors want to apologize for the inclarity. And given one more page, we would like to improve especially on the following aspects: (1) in order to present holistic results across datasets, the splitted tables will either appear on the same page or merged together. We will also avoid discussing Appendix in the main paper; (2) both clustering accuracy and NMI will be presented for all datasets and models in the main results; (3) the paper will be rearranged in a better fashion to include the most important sections into the main paper; (4) the authors will meticulously go over the paper and correct grammatical errors.
>
> > Did you try any other metric for sampling triplets apart from random or entropy-based? The observation the “random sampling inputs for LLM annotations is not optimal/might even hurt performance” seems to be common across multiple works.
>
> Answer: Triplet sampling is crucial in the literature of deep metric learning (DML) [Wang et al, 2014; Schroff et al, 2015; Wu et al, 2017; Xuan et al, 2019]. However, most of their observations are based on the fact that categories (or positive pairs) are already provided. One insightful finding is that using diverse anchors is better than randomly selected anchors [Sumbul et al, 2021]. Thus the authors tried to sample $20$% anchors from each cluster and keep the total number of anchors the same for ClusterLLM, but the results on 3 datasets (Bank77/FewRel/StackEx) did not show significant improvement. Given one more page, we would like to include more discussions on this issue. Additionally, it would be very helpful if you are willing to share several references for the claim “random sampling inputs for LLM annotations is not optimal/might even hurt performance seems to be common across multiple works.”
>
> [Wang et al, 2014] Learning Fine-grained Image Similarity with Deep Ranking
>
> [Schroff et al, 2015] FaceNet: A Uniﬁed Embedding for Face Recognition and Clustering
>
> [Wu et al, 2017] Sampling Matters in Deep Embedding Learning
>
> [Xuan et al, 2019] Improved Embeddings with Easy Positive Triplet Mining
>
> [Sumbul et al, 2021] Informative and Representative Triplet Selection for Multilabel Remote Sensing Image Retrieval
>
> > Spliting main result tables into multiple tables creates a bit of confusion for the reader. I found myself lost in table numbers and flipping pages. Also would be good to mark Supp tables as "Supp. Table 5" to make it explicit.
> >
> > In general i feel the authors can do a better job of presenting the results in a more streamlined fashion with clear agenda contained within the page limits of the paper which wouldn't require the reader to necessarily refer the appendix.
>
> Answer: As stated above, we will present results in a self-contained fashion in order to increase readability.

---

### Official Review · Reviewer_s3g2 · 2023-07-28

**Soundness:** 4

**Excitement:**

4: Strong: This paper deepens the understanding of some phenomenon or lowers the barriers to an existing research direction.

**Paper Topic And Main Contributions:**

The authors present a two step fine-tuning framework that enhances the clustering performance of an embedding model. They optimize smaller models, such as Instructor and E5, using feedback from larger language models like ChatGPT. In the first step, the clusters are refined through the use of triplet queries, asking ChatGPT if A is closer to B than to C. In the second step, a cluster hierarchy is established, by querying whether A and B belong to the same category.

**Questions For The Authors:**

1) The evaluation of the second step (cluster granularity) is rather short. It would have been interesting to see, how well the frameworks perform in the setting when k is unknown. Did you evaluate the clustering accuracy for the models presented in Table 5?

2) Did you perform a qualitative analysis of the merging step and when it stops? Maybe a simple top words per cluster approach would be helpful to visualize, how and why clusters are merged together (or when the merging stops).

**Reasons To Accept:**

1) The paper exhibits a well-structured and compelling organization, effectively presenting its motivation and research content.

2) The authors introduce a novel entropy-based sampling technique to decrease the number of LLM queries needed, making their framework more feasible and efficient.

3) The authors conduct comprehensive experiments demonstrating how their framework outperforms other text clustering methods. The qualitative analysis provides insights into the operational behaviour of the framework.

4) Figures are meaningful and well-designed

**Reasons To Reject:**

Some (minor) points I want to list here:
1) The authors often discuss results presented in the Appendix in the main paper (standalone document).
 Section 4.4 gives an overview of the results, but it would be more beneficial to avoid going into detailed discussions of datasets that are solely listed in the appendix, such as CLINIC(D) and Massive(D). Keeping all crucial information within the main body improves the paper's readability and accessibility. It also seems like the related work section 'Clustering' should be a part of the main paper.

2) Evaluating more than one clustering metric would be vital for analysing the results. The authors only present the NMI results for a subset of models on the last page of the appendix, however, the tables in the main section should also contain NMI in addition to the clustering accuracy.

3) The experiments concerning the second step of the framework (cluster granularity) are relatively brief in length.

**Reproducibility:**

4: Could mostly reproduce the results, but there may be some variation because of sample variance or minor variations in their interpretation of the protocol or method.

**Reviewer Confidence:**

3: Pretty sure, but there's a chance I missed something. Although I have a good feel for this area in general, I did not carefully check the paper's details, e.g., the math, experimental design, or novelty.

**Typos Grammar Style And Presentation Improvements:**

As mentioned earlier, the paper should be self-contained. One should not discuss results presented solely in the appendix in the main section. Tables 5 and 11 seem to be mixed up. Figure 3 is never discussed in Section 4.
Maybe also just a question of style, but it seems the 'one paragraph' Section 2 should be merged into Section 3.

---

> ### Author Rebuttal · Authors · 2023-08-27
>
> We thank you for your comments and feedback. We are glad to see that you notice the novelty of this work is to provide an efficient way to query LLMs and train models. We address your concerns here.
>
> > The authors often discuss results presented in the Appendix in the main paper (standalone document). Section 4.4 gives an overview of the results, but it would be more beneficial to avoid going into detailed discussions of datasets that are solely listed in the appendix, such as CLINIC(D) and Massive(D). Keeping all crucial information within the main body improves the paper's readability and accessibility. It also seems like the related work section 'Clustering' should be a part of the main paper.
>
> Answer: As also pointed out by Reviewer vYLX and Reviewer odh6, the paper presentation needs to be significantly improved in order to increase readability. The authors want to apologize for the inclarity. And given one more page, we would like to improve especially on the following aspects: (1) in order to present holistic results across datasets, the splitted tables will either appear on the same page or merged together. We will also avoid discussing Appendix in the main paper; (2) both clustering accuracy and NMI will be presented for all datasets and models in the main results; (3) the paper will be rearranged in a better fashion to include the most important sections into the main paper; (4) the authors will meticulously go over the paper and correct grammatical errors.
>
> > Evaluating more than one clustering metric would be vital for analysing the results. The authors only present the NMI results for a subset of models on the last page of the appendix, however, the tables in the main section should also contain NMI in addition to the clustering accuracy.
>
> Answer: As stated above, the authors will include both metrics for all datasets and models in the main results.
>
> > The experiments concerning the second step of the framework (cluster granularity) are relatively brief in length.
>
> Answer: As also pointed out by Reviewer vYLX, the proposed method for cluster granularity is under-explored. Reviewer s3g2 suggested providing more analysis. Reviewer vYLX suggested moving these results to appendix or a future work. The authors generally agree that the main focus of this paper is inclined to the first step (perspective of clustering). But we would also emphasize the importance of determining cluster granularity especially for those applications that desire both fine-grained and coarse-grained clustering. Overall, the observation of this paper is that, when presented with in-context examples, LLMs seem to capture the granularity information according to the results on Massive(I/D), MTOP(I/D), CLINC(I/D) in Table.5. However, the authors also agree that more analysis should be done in order to better understand the potential of the method. In particular, we will include visualizations as suggested by Reviewer s3g2 and more ablation studies on the prompts (similar as presented in Appendix.D).
>
> > The evaluation of the second step (cluster granularity) is rather short. It would have been interesting to see, how well the frameworks perform in the setting when k is unknown. Did you evaluate the clustering accuracy for the models presented in Table 5?
>
> Answer: Clustering accuracy is not only dependent on a correct estimation of granularity, but also the clustering quality. An interesting phenomena the authors observed is that, when provided with ground truth number of clusters, the performance may not be maximized for both clustering accuracy and NMI. To give an example, given ground truth number $59$ for Massive(I), the clustering accuracy is $59.35$. However, the clustering accuracy is increased to $63.69$ when provided with the number of clusters $41$ which is smaller than ground truth. The possible reasons could be (1) clustering metrics are sensitive to an imbalanced dataset; (2) clustering quality is not optimal. Hence, the authors prefer to separate the evaluations of clustering quality and clustering granularity estimation in the main results. But we will include these metrics in analysis.
>
> > Did you perform a qualitative analysis of the merging step and when it stops? Maybe a simple top words per cluster approach would be helpful to visualize, how and why clusters are merged together (or when the merging stops).
>
> Answer: This is a good point. These visualizations can potentially help readers understand how LLM predictions look like and will be included in the next version as a case study. Furthermore, we have plotted the consistency score against the number of clusters which shows a continuous curve that peaks at a certain number. We can also observe the range of high scores on this plot and correspond to the keyword result analysis.
>
> > As mentioned earlier, the paper should be self-contained. One should not discuss results presented solely in the appendix in the main section. Tables 5 and 11 seem to be mixed up. Figure 3 is never discussed in Section 4. Maybe also just a question of style, but it seems the 'one paragraph' Section 2 should be merged into Section 3.
>
> Answer: As stated above, the authors will try our best to make the paper self-contained in the next version. We will also add references to Figure 3.

---

### Meta-Review · Area_Chair_pyLE · 2023-09-19

**Recommendation:** 4

**Metareview:**

The reviewers are generally agreed on both their feelings of soundness and excitement, giving high ratings for both categories. The authors are addressing a practical challenge, using methods whose performance will be interesting to many other researchers.

Concerns about clarity recur across the reviews, and this will be important to address for a final version of the paper. Some sections of the paper were confusing to the reviewers and needed additional clarification from the authors during the rebuttal. These confusions were all cleared up, but that additional explanation should be incorporated into the paper.

The reviewers agreed that the experiments were sound and compelling. One reviewer had additional questions about the choice of comparison models, but this was clarified by the rebuttal (and as stated above, these misunderstandings can be easily addressed by clarifying a final version of the paper).

---

### Decision · Program_Chairs · 2023-10-07

**Decision:**

Accept-Main

**Comment:**

The reviewers are generally agreed on both their feelings of soundness and excitement, giving high ratings for both categories. The authors are addressing a practical challenge, using methods whose performance will be interesting to many other researchers.

Concerns about clarity recur across the reviews, and this will be important to address for a final version of the paper. Some sections of the paper were confusing to the reviewers and needed additional clarification from the authors during the rebuttal. These confusions were all cleared up, but that additional explanation should be incorporated into the paper.

The reviewers agreed that the experiments were sound and compelling. One reviewer had additional questions about the choice of comparison models, but this was clarified by the rebuttal (and as stated above, these misunderstandings can be easily addressed by clarifying a final version of the paper).